# Asymmetric synthesis of stereogenic-at-iridium(III) complexes through Pd-catalyzed kinetic resolution

Yun-Peng Chu[1], Xue-Lin Yue[1], De-Hai Liu[1], Chuanyong Wang[2] & Jiajia Ma [1] ✉

Metal-centered chirality has been recognized for over one century, and stereogenic-at-metal complexes where chirality is exclusively attributed to the metal center due to the specific coordination pattern of achiral ligands around the metal ion, has been broadly utilized in diverse areas of natural science. However, synthesis of these molecules remains constrained. Notably, while asymmetric catalysis has played a crucial role in the production of optically active organic molecules, its application to stereogenic-at-metal complexes is less straightforward. In this study, we introduce a kinetic resolution strategy employing a Pd-catalyzed asymmetric Suzuki-Miyaura cross-coupling reaction that efficiently produces optically active stereogenic-at-iridium complexes from racemic mixtures with high selectivity (achieving an *s*-factor of up to 133). This method enables further synthesis of complexes relevant to chiral metallodrugs and photosensitizers, underscoring the practical utility of our approach. Mechanistic studies suggest that reductive elimination is likely the turnover-limiting step over the Suzuki-Miyaura cross-coupling.

Optically active molecules constitute pivotal components in various realms of natural science. Over the past century, significant strides have been made in the development of synthetic methodologies for their acquisition. The pioneering work of synthetic chemists has culminated in the emergence of asymmetric catalysis as a sophisticated and indispensable tool for accessing enantiomerically pure molecules. While the synthetic access to those with chiral main group elements, including carbon, boron, silicon, phosphine, sulfur, and others via asymmetric catalysis, has been successfully established, the pursuit of molecules featuring exclusive metal-centered chirality in a catalytic fashion remains a formidable challenge (Fig. 1a).

Optically active stereogenic-at-metal complexes endowed with exclusive metal-centered chirality have found numerous applications spanning catalysis[1–7], medicinal chemistry[8–12], and material science (Fig. 1b)[13–15]. Despite the elaborated exploration for their preparation since the recognition of metal-centered chirality by Werner in 1900s[16,17], progress has been slow and gradual. Diastereoselective synthesis utilizing enantiomerically pure ligands enables the preparation of chiral transition metal complexes featuring mixed ligand-centered and metal-centered chirality. However, the often non-removability of the chiral ligand renders this approach limited use[18–21]. The state-of-the-art synthetic methods for stereogenic-at-metal complexes can be classified into two main types, as described below (Fig. 1c):

(1) Asymmetric synthesis employing enantiomerically pure anions[22]. For instance, Lacour and colleagues revealed that the chiral phosphate(V) anion TRISPHAT exhibited asymmetric induction in the synthesis of ionic iron(II) complexes[23,24]. Fontecave and coworkers later reported that this strategy was applicable for synthesizing enantiomerically pure ruthenium(II) complexes[25]. However, owing to its reliance on the formation of chiral ion pairs, this strategy is not applicable to obtaining neutral metal complexes.

(2) Asymmetric synthesis utilizing enantiomerically pure auxiliaries[26]. Meggers group has pioneered the chiral auxiliary-mediated asymmetric coordination chemistry[27], and this strategy has found practical applications in synthetic chemistry. For example,

[1]Frontiers Science Center for Transformative Molecules, Shanghai Key Laboratory for Molecular Engineering of Chiral Drugs, School of Chemistry and Chemical Engineering, Zhangjiang Institute for Advanced Study, Shanghai Jiao Tong University, Shanghai, P. R. China. [2]College of Chemistry and Chemical Engineering, Yangzhou University, Yangzhou, China. ✉e-mail: majj@sjtu.edu.cn

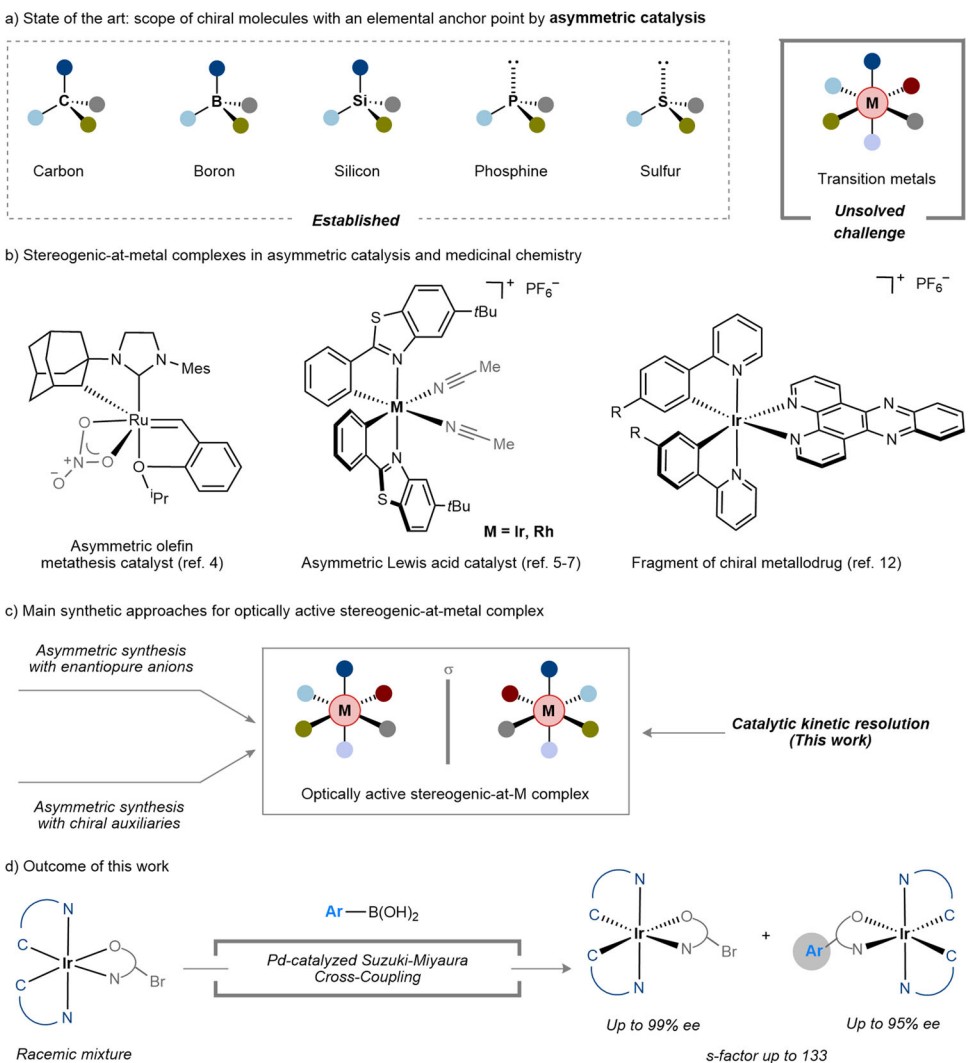

**Fig. 1 | Introduction. a** Molecular chirality. **b** Chiral-at-metal complexes as catalyst and metallodrug. **c** Current main approaches to chiral-at-metal complexes. **d** This work: asymmetric synthesis of chiral-at-iridium complexes through Pd-catalyzed kinetic resolution.

enantiopure homogeneous catalysts[4,5,7], metallodrugs[12] and building blocks of supermolecules[28] can be readily prepared using this strategy.

While these notable advancements, the pursuit of an economical and sustainable synthetic approach, ideally in a catalytic fashion, holds paramount significance. As a seminal contribution in 2010, Meggers and coworkers presented an elegant paradigm wherein optically active [Ru(bpy)$_3$]$^{2+}$ was obtained under catalytic conditions[29], which represents the early instance of utilizing asymmetric catalysis for synthesizing stereogenic-at-metal complexes, thus serving as a proof of concept, albeit with a single example exhibiting 78% ee. Encouraged by this groundbreaking work, we are prompted to explore the potential introduction of alternative catalytic strategies to enrich the synthetic toolbox for optically active stereogenic-at-metal complexes. We herein report the asymmetric transition-metal catalysis is applicable for obtaining octahedral iridium complexes, wherein racemic brominated stereogenic-at-iridium complexes undergo efficient kinetic resolution[30,31] through Pd-catalyzed asymmetric Suzuki-Miyaura cross-coupling (Fig. 1d). Both enantiomers of the stereogenic-at-iridium complexes were obtained with favorable enantioselectivities, thus underscoring the feasibility and utility of this catalytic strategy.

## Results

### Reaction development

Iridium(III) complexes with two identical cyclometalating ligands and a second complementary one are the versatile scaffold as photocatalyst[5,32–36], metallodrug[10,12,37], organic-light-emitting-diode (OLED) emitter[38–42] etc. As such, given their significance in these fields, cyclometalated iridium(III) complexes were chosen as the objective of this study. Experimentally, inspired by the seminal asymmetric Suzuki-Miyaura cross-coupling works by Buchwald[43], Cammidge[44], and many others[45–53], we commenced the study by examining the reaction of racemic brominated iridium(III) complex *rac*-**1a** and a boronic acid **2a** under palladium-catalyzed conditions (Table 1). The commercial bidentate chiral phosphine ligands (L1–L3) and Wang's bridged cyclic phosphine ligand L4[54] were unable to facilitate the reaction (entry 1). Subsequently, various types of phosphonamidite ligands were explored, Feringa's phosphonamidite L7 was found to be applicable for the kinetic resolution of *rac*-**1a**, thus giving the desired cross-coupling product Δ-**3a** with 59% ee (entry 2). The partially saturated Feringa's ligand L8 led to an increased enantioselectivity for Δ-**3a** (entry 3). Tuning the electronic property of the chiral benzylic amines sphere revealed that the electron-rich derivative (L9) was superior over the electron-deficient one (L10) in terms of the kinetic resolution selectivity (entries 4 versus 5). However,

## Table 1 | Reaction optimization [a]

L8  $R^1 = R^2 = Ph$
L9  $R^1 = R^2 = 4\text{-MeOC}_6H_4$
L10 $R^1 = R^2 = 4\text{-FC}_6H_4$
L11 $R^1 = 3,4,5\text{-MeO}_3C_6H_2$
      $R^2 = 4\text{-MeOC}_6H_4$

| Entry | Z | T (°C) | Ligand | t | ee of Λ-1 [b] | ee of Δ-3 [b] | C (%) [c] | s [d] |
|---|---|---|---|---|---|---|---|---|
| 1 | H (1a) | 50 | L1–L6 | 24 h | – | – | NR | – |
| 2 | H (1a) | 50 | L7 | 12 h | 40 | 59 (3a) | 40 | 5.7 |
| 3 | H (1a) | 50 | L8 | 12 h | 23 | 65 (3a) | 26 | 5.9 |
| 4 | H (1a) | 50 | L9 | 12 h | 16 | 69 (3a) | 19 | 6.4 |
| 5 | H (1a) | 50 | L10 | 12 h | 37 | 17 (3a) | 69 | 1.9 |
| 6 | H (1a) | 50 | L11 | 12 h | 8 | 59 (3a) | 12 | 4.2 |
| 7 | Me (1b) | 30 | L9 | 12 h | 9 | 87 (3b) | 9 | 15.7 |
| 8 | MeO (1c) | 30 | L9 | 12 h | 9 | 96 (3c) | 9 | 54 |
| 9 | iPrO (1d) | 30 | L9 | 12 h | 44 | 87 (3d) | 34 | 22 |
| 10 [e] | MeO (1c) | 30 | L9 | 3.5 d | 90 | 90 (3c) | 50 | 58 |

[a]Conditions: Rac-1 (0.01 mmol), 2a (0.02 mmol), Pd$_2$dba$_3$ (5 mol%, 0.0005 mmol), Ligand (12 mol%, 0.0012 mmol) and CsF (0.02 mmol) in THF/H$_2$O (v/v = 9/1, 0.1 mL) stirred at designated temperatures under N$_2$.
[b]Determined by HPLC analysis on a chiral stationary phase, and the absolute values are displayed.
[c]Conversion (C) = ee$_S$/(ee$_S$+ee$_P$).
[d]s = ln[(1−C)(1−ee$_S$)]/ln[(1−C)(1+ee$_S$)].
[e]Reaction was performed on a 0.05 mmol scale. For more details, see Supplementary Information Reaction Optimization.

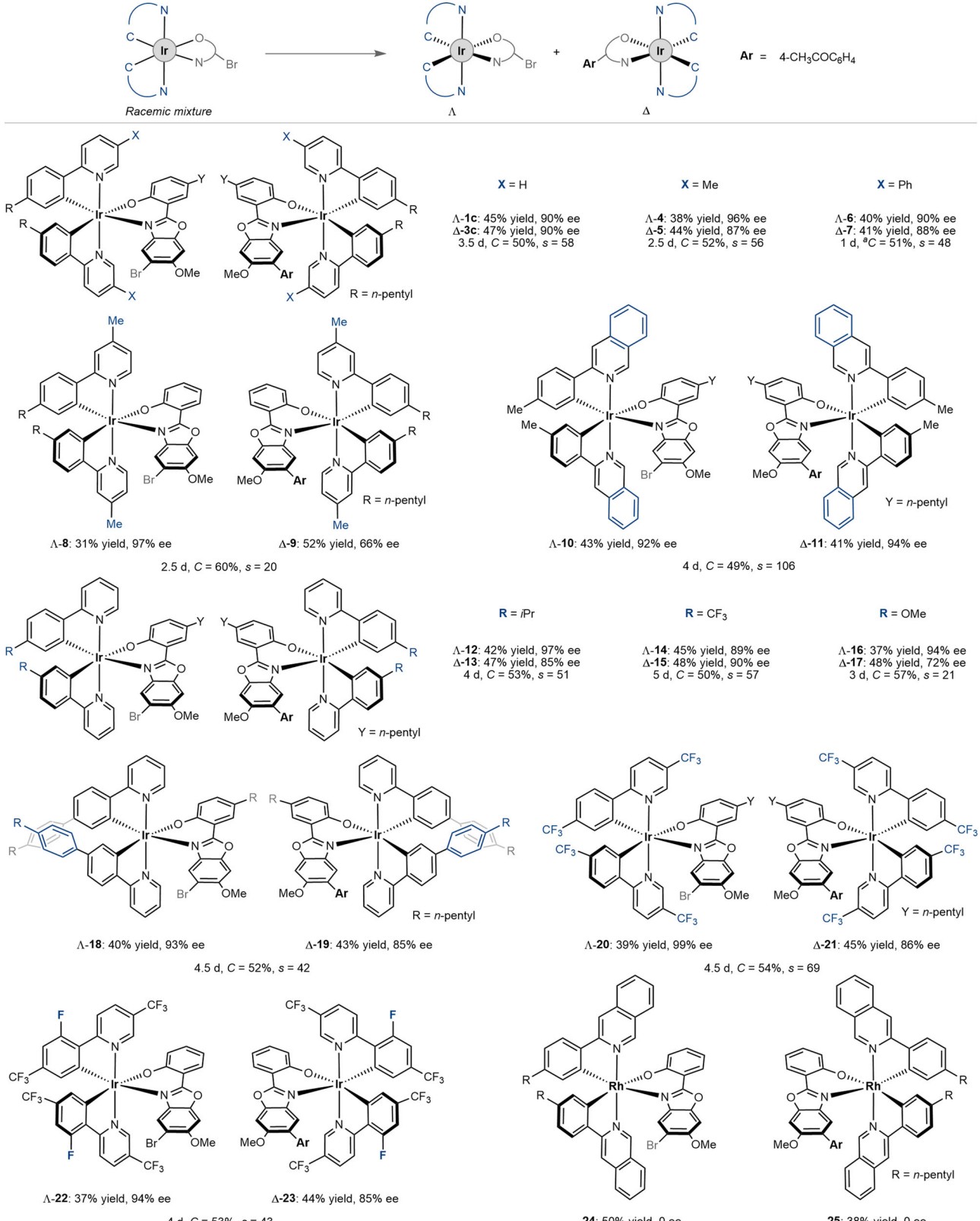

**Fig. 2 | Scope of iridium(III) complexes.** Conditions: *Racemic mixture* (0.05 mmol), **2a** (0.10 mmol), Pd₂dba₃ (5 mol%, 0.0025 mmol), ligand (12 mol%, 0.0060 mmol) and CsF (0.10 mmol) in THF/H₂O (v/v = 9/1, 0.5 mL) stirred at 30 °C. *ᵃ*The reaction was conducted at 35 °C.

introducing more electron-donating methoxys (L11 versus L9) led to inferior results (entry 6). Next, we moved to adjust the substrate by introducing substituents to the ortho position of the bromo in *rac-*1a. To our delight, with a methyl (Z = Me), this Pd-catalyzed Suzuki-Miyaura reaction turned out to be operative under decreased temperature (50 → 30 °C), which has been recognized as a key parameter for obtaining high selectivity[42], thus giving an increased *s*-factor of 15.7 (entry 7). Further, a methoxy (Z = MeO) was found to be optimal with the highest *s*-factor whilst a bulkier *ⁱ*PrO led to inferior results (entries 8 versus 9). A scale-up reaction with prolonged reaction time

furnished the cross-coupling product Δ-**3c** with 90% ee under the recovery of Λ-**1c** with 90% ee, thus corresponding to a conversion of 50% and an *s*-factor of 58 (entry 10).

## Scope and generality evaluation

With the optimal kinetic resolution conditions in hand, we next evaluated the scope of racemic iridium(III) complexes and the cross-coupling partners. As outlined in Fig. 2, placing a methyl or phenyl group at the 5-position of pyridyl, upon kinetic resolution, led to the cross-coupling products Δ-**5** and Δ-**7** in 87-88% ee under the recovery of Λ-**4** and Λ-**6** in 90−96% ee. Even though 4-methyl pyridyl gave the cross-coupling product Δ-**9** in 66% ee, the remained Λ-**8** was obtained with excellent enantioselectivity of 97%. Interestingly, an extended ring system of isoquinoline led to product Δ-**11** in 94% ee under the recovery of Λ-**10** in 92% ee, corresponding to a conversion of 49% and an excellent *s*-factor of 106. Next, we evaluated the substituent effect on cyclometalating phenyl ring. A bulkier *i*Pr (Δ-**13**), an electron-deficient CF₃ (Δ-**15**), an electron-donating MeO (Δ-**17**), and a phenyl (Δ-**19**) are compatible under this kinetic resolution protocol, thus giving the *s*-factor of 21−57. Further, bis-trifluoromethylated racemic iridium(III) complexes **20** and **22**, which often serve as photocatalyst scaffolds, can also undergo the Pd-catalyzed kinetic resolution to give optically active stereogenic-at-iridium complexes with up to 99% ee. It's noteworthy that while replacing the Ir with Rh (**24**), the kinetic resolution became not operative in which both the cross-coupling product **25** and the remaining **24** were found to be racemic at a conversion of 43%. A kinetic study was conducted by mixing rhodium complex **25**, and the aryl bromide ligand **26** in CDCl₃ (Fig. 3). After 5 min, ¹H NMR analysis of the mixture revealed dissociation of ionic **27** (29% yield) from rhodium center as well as the production of complex **24** (32% yield). These observations suggested a rapid ligand exchange kinetic, which indicates that over our attempt for kinetic resolution of rhodium complex **24**, the Suzuki-Miyaura cross-coupling might take place on the dissociated aryl bromide ligand **26** to afford **27**. The following re-coordination of **27**, upon deprotonation, delivered complex **25**. Therefore, chiral recognition between

substrate **24** and catalyst [Pd/L9] is not operative in this case (for more details, see Supplementary Information Study of Coordination Stability of Rh-25). The scope of the coupling partners was further evaluated (Fig. 4). Accordingly, aryl boronic acid, the corresponding pinacol ester (Bpin), and trifluoroborate potassium salt led to comparable results (Λ-**28** & Δ-**29**) while the trifluoroborate potassium salt required an elevated reaction temperature of 40 °C for obtaining high conversion. The acetyl substituent at both *meta* (Δ-**30**) and *ortho* (Δ-**31**) positions of phenyl were compatible. Due to the presence of two substituents at the biaryl of Δ-**31**, rotational isomerization was observed, as suggested by HPLC analysis and room/high-temperature ¹H NMR analysis (see Supplementary Information High-Temperature NMR Experiment of Compound 31). Besides these, other electron-withdrawing groups, such as ester and sulfonyl led to cross-coupling products in excellent ee of 95% (Δ-**32**) and 92% (Δ-**33**), respectively. Meanwhile, electron-rich phenyl boronic acid derivatives are also suitable coupling partners thus giving Δ-**34**−**36** in 91−93% ee. Interestingly, heteroaromatic and olefinic boronic acids (products Δ-**37**−**39**) are suitable coupling partners in this Pd-catalyzed kinetic resolution protocol. An acid-induced ligand exchange reaction of Δ-**11** was conducted in the presence of 2,2′-bipyridine, thus providing complex **40**, which is suitable for crystallization (Fig. 5). X-ray crystallographic analysis of **40** ambiguously indicated the Δ configuration of complex **11**. Circular dichroism (CD) analysis showcased the opposite configuration of complexes **10** and **11**. Therefore, the former one was assigned as Λ (Fig. 4b). Configuration of other complex pairs, such as **12** vs. **13**, **14** vs. **15**, etc., were further assigned accordingly by CD analysis (see Supplementary Information Circular Dichroism).

## Synthetic application

Next, the brominated iridium complex Λ-**28** can undergo diverse synthetic transformations (Fig. 6). Under achiral Pd(PPh₃)₄-catalyzed cross-coupling or hydrodehalogenation conditions, Λ-**28** was smoothly converted into complexes **41** and **42**, respectively, in excellent enantioselectivities of >99% and 99%. Meanwhile, acid-induced

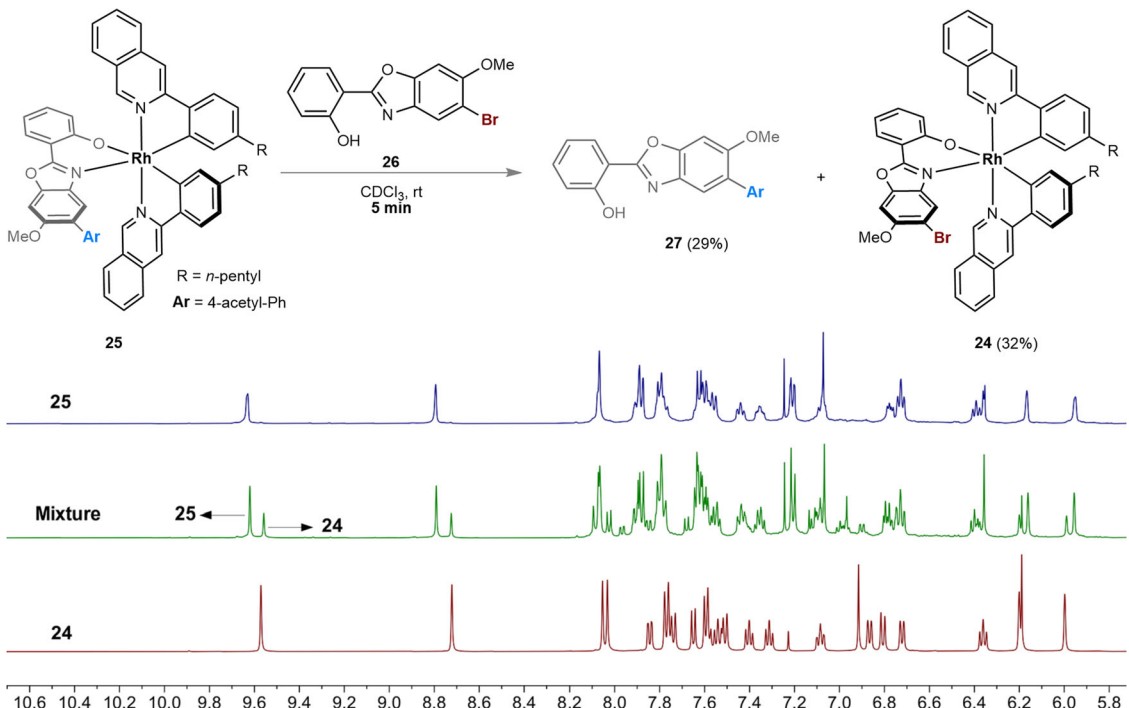

**Fig. 3 | Probing rapid ligand exchange kinetic of rhodium complex 25 using ¹H NMR spectrum.** For more details, see Supplementary Information Study of Coordination Stability of **25**.

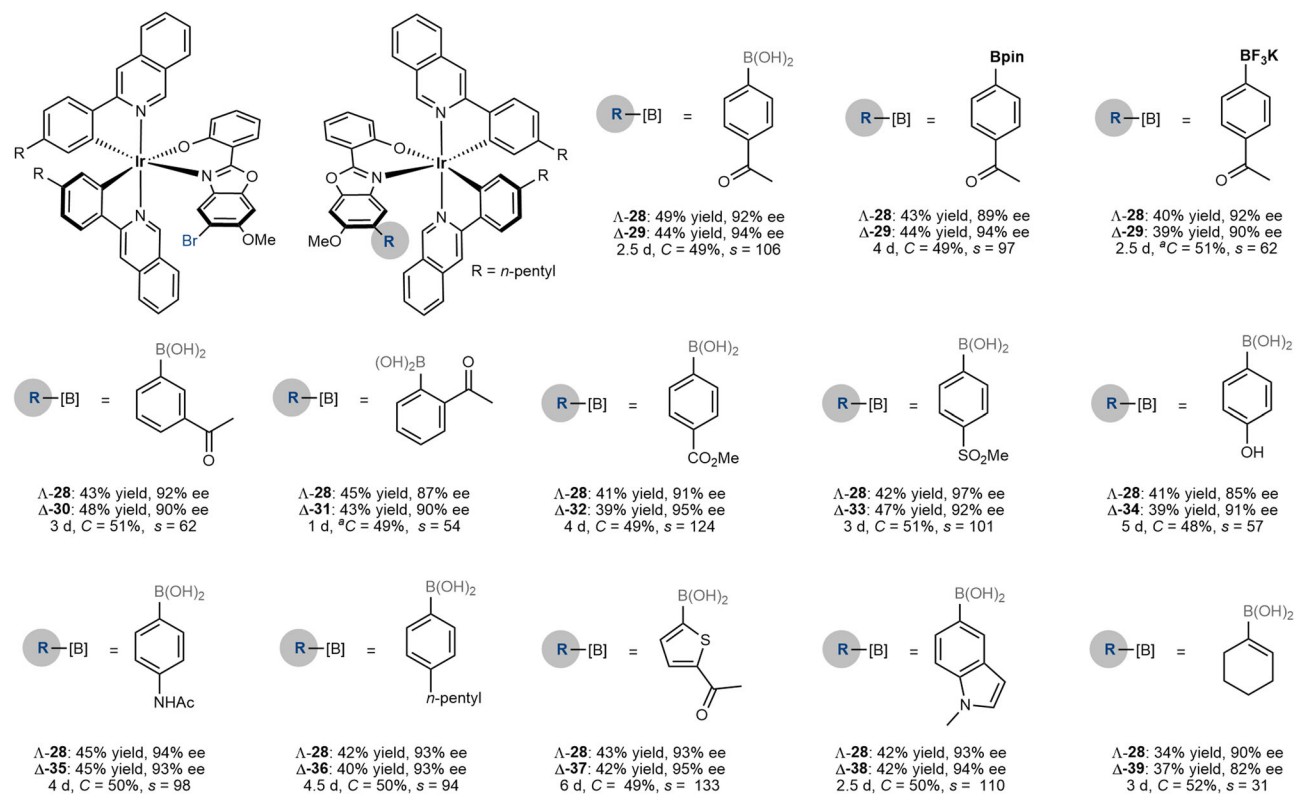

**Fig. 4 | Scope of cross-coupling partners.** Conditions: *Rac*-**1** (0.05 mmol), **2** (0.10 mmol), Pd$_2$dba$_3$ (5 mol%, 0.0025 mmol), ligand (12 mol%, 0.0060 mmol) and CsF (0.10 mmol) in THF/H$_2$O (v/v = 9/1, 0.5 mL) stirred at 30 °C. *$^a$The reaction was conducted at 40 °C.

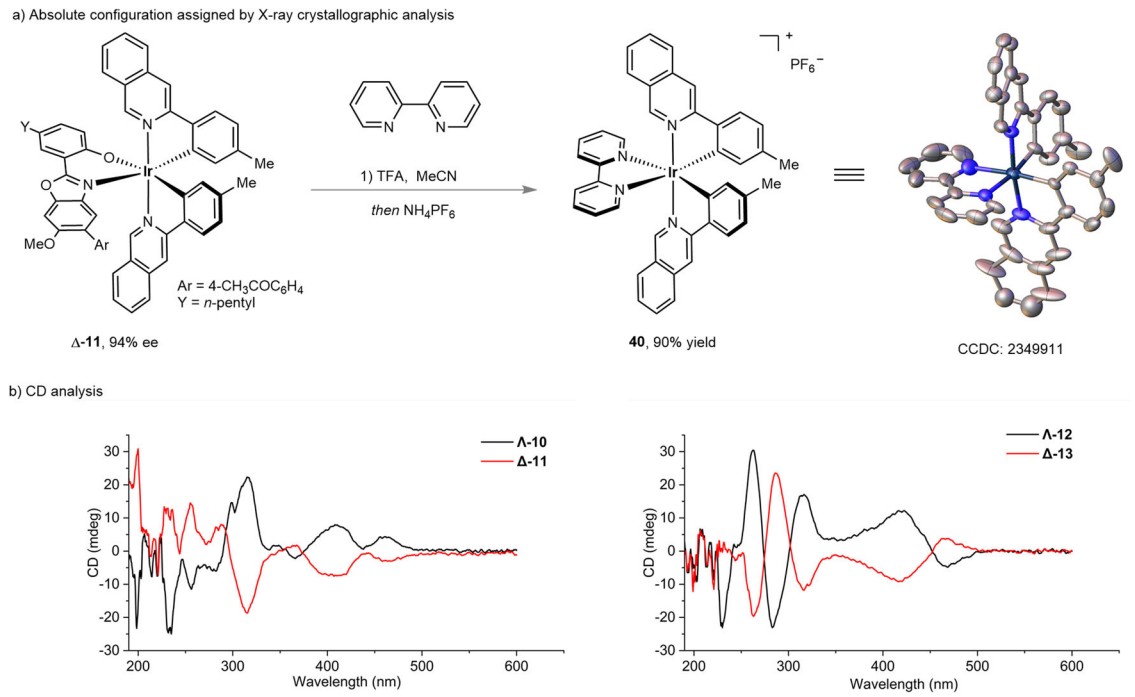

**Fig. 5 | Absolute configuration assignment. a** X-ray crystallographic analysis. **b** CD analysis with spectrum recorded in MeCN (0.000025 mol/L).

ligand exchange reactions with acetylacetone or a phenanthroline derivative were performed to furnish complexes **43** and **44**, respectively, which are related scaffolds of OLED emitter[38] and metallodrug[12]. The acid-induced N,O-bidentate ligand cleavage in CH$_3$CN gave the hemisolvated complex **45**, which is a versatile intermediate in chiral iridium complexes synthesis. Notably, the chiral photosensitizer **46**, as

demonstrated by Yoon and coworkers[55] was obtained in 90% yield by ligand exchange of Λ-**20** with 2-(1*H*-pyrazol-3-yl)pyridine.

## Mechanistic investigation

To probe the turnover-limiting step in the Pd-catalyzed Suzuki-Miyaura cross-coupling reaction, kinetic investigations were

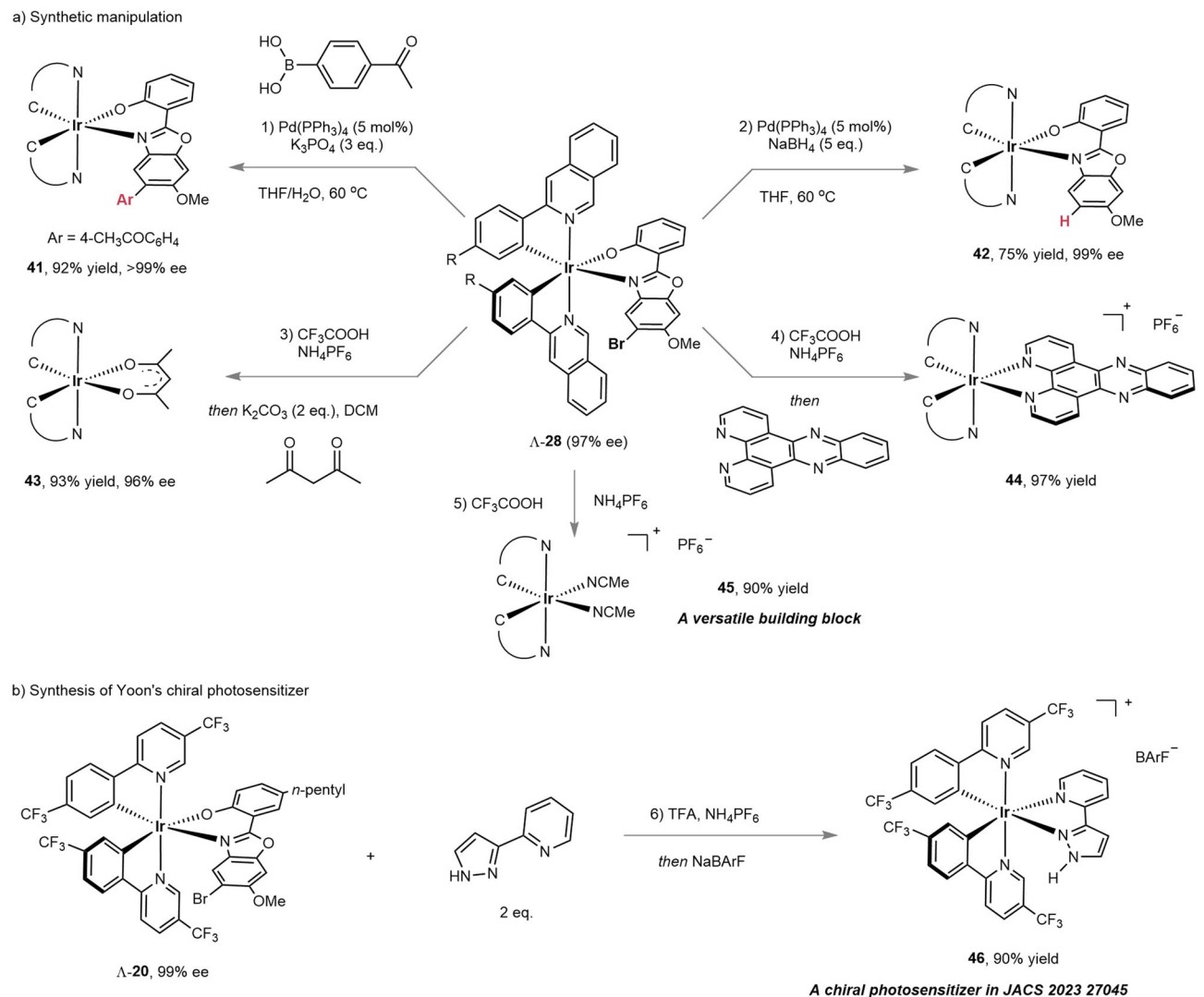

**Fig. 6 | Synthetic application. a** Synthetic manipulations. **b** Synthesis of Yoon's chiral photosensitizer. For more reaction details, see Supplementary Information Synthetic Transformations.

performed (Fig. 7). Initial reaction rates were measured under an array of concentrations regarding racemic brominated iridium complex **28**, boronic acid **2a** and the precatalyst [Pd/L9], respectively. As a result, increasing the concentration of **28** or **2a** didn't affect the initial reaction rates which is consistent with a zeroth-order dependence (Fig. 7a, b). Therefore, neither the oxidative addition nor transmetalation is likely turnover-limiting. Meanwhile, this cross-coupling reaction was found as first-order independence on the precatalyst [Pd/L9], thus suggesting a monomeric Pd complex engaging in the turnover-limiting step (Fig. 7c). Next, identical conversions of 19% were observed while subjecting boronic acid (**2a**) and borates (**2b** and **2m**) to the standard reaction conditions, respectively (Fig. 7d). This observation further excludes out that transmetalation is the turnover-limiting step. Last, the steric effect at the ortho position of Br was evaluated (Fig. 7e). Accordingly, the cross-coupling product was not detected at 30 °C with **1a** (Z = H) while placing a methyl (**1b**, Z = Me) or methoxy group (**1c**, Z = MeO) led to a remarkable conversion of 9%. A bulkier $^i$PrO was capable of accelerating the reaction significantly with a higher conversion of 34%. Therefore, the reaction rate correlates with the size of substituents at the ortho position of Br. Collectively, these kinetic studies indicate that reductive elimination might serve as the

turnover-limiting step. Nevertheless, given the selectivity is controlled by the rate of formation of the Δ and Λ oxidation adducts, we still hypothesize that oxidative addition is likely the selectivity-determining step.

We herein demonstrate optically active stereogenic-at-iridium complexes can be obtained by kinetic resolution under a Pd-catalyzed asymmetric Suzuki-Miyaura cross-coupling reaction. Excellent enantioselectivities of up to 99% ee and high kinetic resolution performance with s-factor of up to 133 were obtained. This work remarkably complements the current synthetic toolbox for stereogenic-at-metal complexes and we anticipate that it'll spur more advancements for accessing stereogenic-at-metal complexes with asymmetric catalysis.

## Methods

### General procedure for the synthesis of stereogenic-at-iridium(III) complexes

A dried 10 mL Schlenk tube was charged with the *rac*-**Ir** (0.05 mmol, 1.0 equiv.), arylboronic acid (0.1 mmol, 2 equiv.), Pd$_2$(dba)$_3$ (0.0025 mmol, 5 mol%), L9 (0.006 mmol, 12 mol%), CsF (0.1 mmol, 2 equiv.) and THF/H$_2$O (v/v, 9:1, 0.5 mL) under N$_2$. The mixture was stirred at 30 °C, and the conversion was monitored by HPLC analysis. After achieving the desired

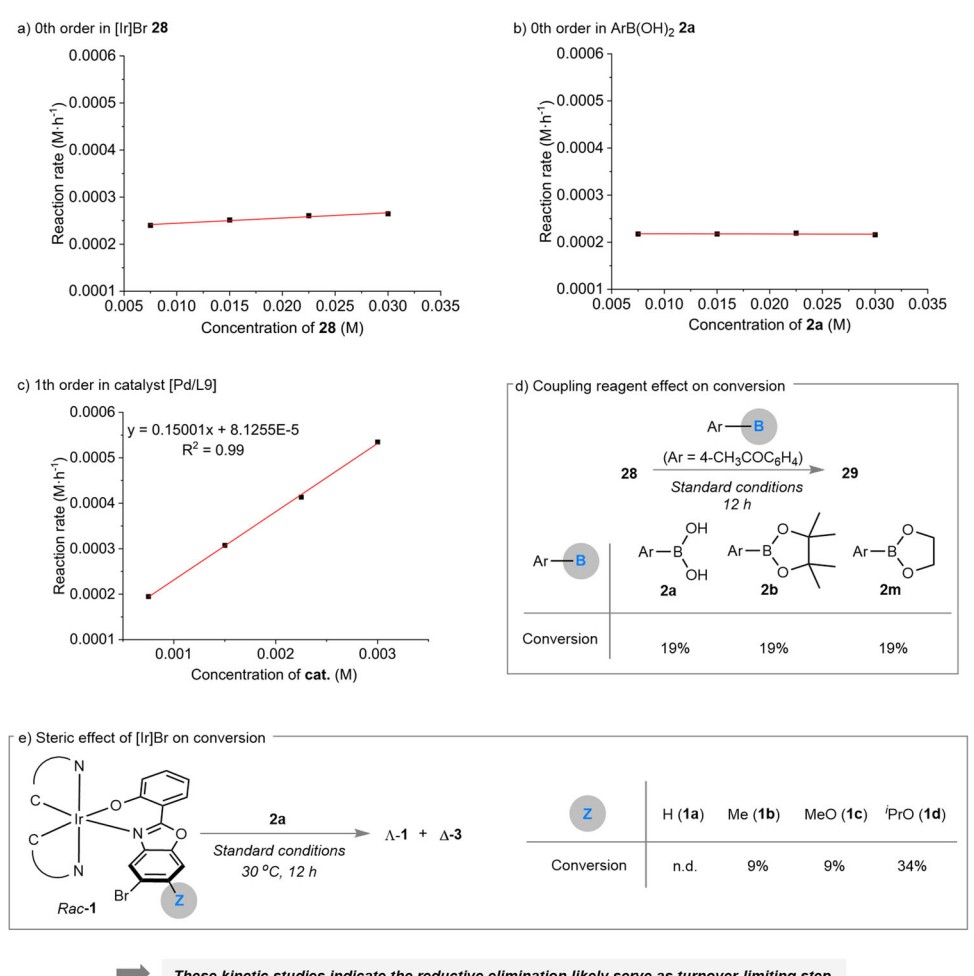

**Fig. 7 | Mechanistic investigation. a** 0th order in [Ir]Br **28**. **b** 0th order in ArB(OH)$_2$ **2a**. **c** 1th order in catalyst [Pd/L9]. **d** Coupling reagent effect on conversion. **e** Steric effect of [Ir]Br on conversion.

conversion, the mixture was concentrated under vacuum, yielding a residue that was purified via column chromatography on silica gel, affording the recovered Λ-**Ir** and the cross-coupling product Δ-**Ir**.

## Data availability
All data supporting the findings of this study are available within the article and its Supplementary Information files or from the corresponding author upon request. Crystallographic data for the structure reported in this article have been deposited at the Cambridge Crystallographic Data Center, under deposition number 2349911 for compound **40**. Copies of the data can be obtained free of charge via https://www.ccdc.cam. ac.uk/structures/.

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

## Acknowledgements

J.M. and Y.C. thank Prof. Mouhai Shu and Gang Chen (all SJTU) for their great generosity for sharing labs and instruments. National Key R&D Program of China (2023YFA1508900) and National Nature Science Foundation of China (22201174) are gratefully acknowledged for financial support.

## Author contributions

J.M. conceived and supervised the project. Y.-P.C., X.-L.Y., and D.-H.L. designed and performed the synthetic experiments. C.W. analyzed the single crystal structures. J.M. wrote the manuscript with contributions from all authors.

## Competing interests

The authors declare no competing interest.
