## [Transparent Peer Review file · Nature Communications]

Asymmetric Synthesis of Stereogenic-at-Iridium(III) Complexes through Pd-Catalyzed Kinetic Resolution

Corresponding Author: Professor Jiajia Ma

Version 0:

Reviewer comments:

Reviewer #1

(Remarks to the Author)

This manuscript by Ma and coworkers reveals a new methodology for asymmetric synthesis of stereogenic-at-metal complexes. Considering the recent expanding interest in metal-centered chirality and the limited available methods for their synthesis, this topic is timely and of high importance. The methodology that has been developed is based on the kinetic resolution strategy via Suzuki-Miyaura coupling. The Suzuki-Miyaura coupling can be done using quite simple reaction conditions and high selectivity (s -factor up to 133) may be reached at room temperature, allowing access to a series of enantio-enriched chiral-at-Ir complexes. Regarding the originality of this paper and the good results obtained, I support the publication of this article, after addressing several minor comments.

(1) Different from the classical strategy using chiral auxiliaries to form separable diastereomers, this kinetic resolution strategy leads to the formation of the coupling product and the remaining starting material. From a practical aspect of view, it would be ideal that the coupling product is easily separated from the starting material. The authors may comment on this or provide the detailed separating conditions in the SI.

(2) Review references concerning kinetic resolution should be cited.

(3) The coupling reaction takes place on the N,O-bidentate ligand, and the scope of different arylboronic acids was studied in Figure 4. However, it seems the resulting N,O-ligand part should be removed for further applications as demonstrated in Figure 6, which makes this scope study less attractive. I believe that the usefulness of this method will be further enhanced if direct application of the coupling products is demonstrated, if possible.

(4) The practical use of the stereogenic-at-metal complexes usually requires an excellent ee like >99%, is it possible to further enrich the ee's of the products, for example by recrystallization?

(5) Line 144 in main text, complex 26 should be complex 28; SI-page S77, the peak corresponding to the major enantiomer contains significant amount of impurity in the HPLC chart.

Reviewer #2

(Remarks to the Author)

In this manuscript, Ma and co-workers present a clever and elegant strategy to synthesize chiral-at-iridium complexes by kinetic resolution (kr) of the corresponding brominated racemic precursors through Pd-catalyzed asymmetric Suzuki-Miyaura cross-coupling. From a practical point of view, it is questionable whether this approach (requiring 10 mol% of a Pd source, 12 mol% of a chiral phosphoramidite ligand, manipulation in a glovebox, and column chromatography for purification) is comparable to other methodologies that allow for a straightforward synthesis of many chiral-at-iridium complexes (for instance the use of readily available resolution agents like L-proline or L-serine, following the well-established methodologies developed by Meggers). Nevertheless, there is no doubt that, conceptually, this catalytic approach is extraordinarily relevant. Additionally, it widens the available tools for the resolution of chiral-at-metal compounds, and it can become the Holy Grail for the resolution of other stereogenic-at-metal compounds for which the "classical" methodologies are not applicable, provided that the central core is configurationally stable.

The authors have demonstrated the feasibility of the approach and its practical application for synthesizing a wide variety of derivatives. For this purpose, they have explored the reactivity of the formed product and the unreacted brominated substrate (both obtained as enantioenriched samples after kr), showing that a broad range of synthetic transformations are possible. Considering the novelty and relevance of the strategy and its potential impact in the field, I strongly support its publication in Nature Communications.

Nevertheless, there are some suggestions and minor corrections that could help the authors improve the quality of the work. Suggestions:

- The work presented is thoroughly supported by many experimental evidences, and there is no doubt that kr occurs with more than acceptable selectivities on a wide range of substrates and with different coupling partners. Nevertheless, it would be nice to include some discussion or hypothesis on the origin of the observed selectivities. In the last section of the manuscript, the authors conduct some kinetic experiments to define the rate-determining step of the process. Since the reaction is run under kinetic resolution conditions (selectivity being controlled by the rate of formation of δ and λ Suzuki-Miyaura adducts) and the authors identify reductive elimination as rds, do they assume that reductive elimination is also the selectivity-determining step?
- Authors claim that the reaction is catalyzed by a monometallic Pd complex (presumably containing one chiral phosphoramidite ligand according to the P/Pd ratio used). Did the authors consider performing some theoretical calculations to evaluate the relative energy of the different diastereomeric reaction intermediates and transition states?
- The preliminary optimization process, explained in detail in the Supplementary Information, should be mentioned in the main text. Actually, looking at the information presented in the SI, the authors explored a much larger number of chiral ligands than those shown in Table 1 (though not the same labeling scheme is used). Surprisingly, the discussion about the influence of the ligand properties on the effectivity of the kr in the main text is limited to a very simple comment on the effect of some electronic properties. Considering that the best results in terms of kr have been obtained with phosphoramidite ligands displaying an atropisomeric binaphthyl (or related) unit and stereogenic carbon centers, and some diastereomers of the same ligand have been explored (i.e. L7(=LS17) and LS25), it would be interesting to analyze/discuss if the enantioselection is mainly dictated by the chiral descriptor of the binaphthyl unit or by the stereogenic carbon atoms and which are the matching/mismatching combinations of both chiral elements. For this discussion, it would be necessary to specify the chiral descriptor (δ or λ) of the major enantiomer of the Suzuki-Miyaura adduct formed. Could the authors consider incorporating this information (if available) to Figures S1-S3 and Table 1?
- In this vein, the authors decided to label the residual unreacted substrate λ and the Suzuki-Miyaura adducts δ , suggesting that these are the main isomers of the unreacted and formed complexes, respectively. This descriptor is confirmed in the case of compounds δ -11 and λ -10, by analysis of the X-ray structure of derivative 40. It is stated that other complex pairs were further assigned accordingly by CD analysis. It seems unclear how these assignments were performed. Did the authors assume that with ligand L9 the main adduct formed was δ -configured regardless the organometallic core and arylboronic acid used? Why only a selection of compounds is analyzed by CD?

Minor corrections/typo mistakes:

- Please specify the reaction conditions in Figures 2 and 4.
- In Figure 4 the superscript a is used to indicate that the reaction with ArBF_3K was performed at 40 °C. Please use the same label for the reaction forming product 31. Since all the reactions were conducted to close to 50% conversions, please specify the reaction time for each entry.
- In page 7 (line 121 in pdf). Please, rephrase: "Due to the exist"
- In Figure 5 the n-pentyl substituent in δ -11 is missing.
- In the kinetic experiments, please specify the concentrations of $\text{Pd}(\text{dba})_3$ and ligand used. In the Supplementary Information (page S181) it reads "concentrations of cat. used: 0.00075 M, 0.0015 M, 0.00225 M, 0.003 M."
- In the kinetic study, one of the aryl-borates is labeled as 2m in the text and in Figure 7d. This labeling is unclear as this coupling partner was not included in Figure 4. Additionally, in Figure 4 the borates were not labeled.

Reviewer #3

(Remarks to the Author)

In this manuscript, Ma and coworkers disclosed the asymmetric synthesis of stereogenic-at-metal complexes, through an innovative kinetic resolution strategy under a Pd-catalyzed asymmetric Suzuki-Miyaura cross-coupling reaction. Conventionally, the synthesis of stereogenic-at-metal complexes would rely on stoichiometric amounts of chiral reagents, auxiliaries or counterions, which indeed works, but represents very old-school methods, encountering with low step- and atom economies. Asymmetric catalysis, an ideal synthetic tool, if works, would undoubtedly triggers the revolution in the field. As such, this work is significantly notable for its innovation, as well as the obtained high selectivity (mostly >90% ee and s factor >50) in producing optically active stereogenic-at-iridium complexes from racemic mixtures. Further, the authors also demonstrated the synthetic diversification of the obtained complexes, showing the synthetic routes to metallodrugs and chiral photocatalysts. A solid kinetic investigation was ultimately performed, suggesting that the reductive elimination is the rate-determining step over Suzuki coupling reactions. The authors have contributed a thorough and robust work in this manuscript.

Overall, this reviewer would like to suggest the publication of this work in Nature Communications after the following issues are addressed.

- 1) Please rephrase the title in which "stereogenic-at-metal" might be more specific and acceptable than molecules featuring metal-centered chirality.
- 2) It's interesting to know the Rh-complex is more kinetically unstable. The authors may emphasize the configurational stability of Ir congeners towards heat, moisture, various solvents. Would be nice to know for any users.
- 3) Figure 6, the numbering of reaction conditions should start from 1) but not 2).
- 4) Please double check the minus symbol for compound 43 as well as others.
- 5) Sample concentration of the CD analysis must be provided in the figure legend.

Version 1:

Reviewer comments:

Reviewer #1

(Remarks to the Author)

The revised version has addressed most of my concerns. It is now recommended to be published in nature communications.

Reviewer #2

(Remarks to the Author)

In this manuscript, Ma and co-workers present a clever and elegant strategy to synthesize chiral-at-iridium complexes by kinetic resolution (kr) of the corresponding brominated racemic precursors through Pd-catalyzed asymmetric Suzuki-Miyaura cross-coupling. From a practical point of view, it is questionable whether this approach (requiring 10 mol% of a Pd source, 12 mol% of a chiral phosphoramidite ligand, manipulation in a glovebox, and column chromatography for purification) is comparable to other methodologies that allow for a straightforward synthesis of many chiral-at-iridium complexes (for instance the use of readily available resolution agents like L-proline or L-serine, following the well-established methodologies developed by Meggers). Nevertheless, there is no doubt that, conceptually, this catalytic approach is extraordinarily relevant. Additionally, it widens the available tools for the resolution of chiral-at-metal compounds, and it can become the Holy Grail for the resolution of other stereogenic-at-metal compounds for which the "classical" methodologies are not applicable, provided that the central core is configurationally stable.

The authors have demonstrated the feasibility of the approach and its practical application for synthesizing a wide variety of derivatives. For this purpose, they have explored the reactivity of the formed product and the unreacted brominated substrate (both obtained as enantioenriched samples after kr), showing that a broad range of synthetic transformations are possible. Considering the novelty and relevance of the strategy and its potential impact in the field, I strongly support its publication in Nature Communications.

Since the authors have appropriately addressed some minor suggestions/corrections mentioned in my previous report, I recommend accepting it for publication in its current form.

Reviewer #3

(Remarks to the Author)

I think this highly valuable revised manuscript is suitable for the publication in Nature Communications.

made.

Reviewer #1 (Remarks to the Author):

This manuscript by Ma and coworkers reveals a new methodology for asymmetric synthesis of stereogenic-at-metal complexes. Considering the recent expanding interest in metal-centered chirality and the limited available methods for their synthesis, this topic is timely and of high importance. The methodology that has been developed is based on the kinetic resolution strategy via Suzuki-Miyaura coupling. The Suzuki-Miyaura coupling can be done using quite simple reaction conditions and high selectivity (s-factor up to 133) may be reached at room temperature, allowing access to a series of enantio-enriched chiral-at-Ir complexes. Regarding the originality of this paper and the good results obtained, I support the publication of this article, after addressing several minor comments.

Thank the reviewer for their positive assessment and detailed feedback. We appreciate the recognition of the novelty of our methodology in the asymmetric synthesis of stereogenic-at-metal complexes. We have carefully addressed the minor issues and ensured all points have been resolved to your satisfaction.

(1) Different from the classical strategy using chiral auxiliaries to form separable diastereomers, this kinetic resolution strategy leads to the formation of the coupling product and the remaining starting material. From a practical aspect of view, it would be ideal that the coupling product is easily separated from the starting material. The authors may comment on this or provide the detailed separating conditions in the SI.

Response: As suggested, we have included detailed separating conditions in the revised supplementary information.

(2) Review references concerning kinetic resolution should be cited.

Response: As suggested, we have included pertinent references concerning kinetic resolution in manuscript.

“We herein report the asymmetric transition-metal catalysis is applicable for obtaining octahedral iridium complexes, wherein racemic brominated stereogenic-at-iridium complexes undergo efficient kinetic resolution^{30,31} through Pd-catalyzed asymmetric Suzuki-Miyaura cross-coupling (Figure 1d).”

[30] Vedejs, E. & Jure, M. Efficiency in Nonenzymatic Kinetic Resolution. *Angewandte Chemie International Edition* **44**, 3974-4001, (2005).

[31] Pellissier, H. Catalytic Non-Enzymatic Kinetic Resolution. *Advanced Synthesis & Catalysis* **353**, 1613-1666, (2011).

(3) The coupling reaction takes place on the N,O-bidentate ligand, and the scope of different arylboronic acids was studied in Figure 4. However, it seems the resulting N,O-ligand part should be removed for further applications as demonstrated in Figure 6, which makes this scope study less attractive. I believe that the usefulness of this method will be further enhanced if direct application of the coupling products is demonstrated, if possible.

Response: Thank you very much for the valuable feedback and constructive comments. We appreciate your insights regarding the scope of the coupling reaction and the potential for direct application of the coupling products. Regarding your suggestion to demonstrate the direct

application of the coupling products, we understand the importance of showing their practical utility. However, at this stage, due to resource and technical constraints, we are unable to fully explore and demonstrate these applications. We are committed to further investigating the functional aspects of the coupling products, particularly in the context of metal-based drugs, and aim to include such results in future publications.

(4) The practical use of the stereogenic-at-metal complexes usually requires an excellent ee like >99%, is it possible to further enrich the ee's of the products, for example by recrystallization?

Response: Thank the reviewer for the insightful comments. We fully agree with the point that high enantiomeric excess (ee), often greater than 99%, is crucial for the practical application of stereogenic-at-metal complexes. We have indeed considered using recrystallization to further enrich the ee of the products, as suggested. Our preliminary studies have explored various solvent systems and crystallization conditions to assess their impact on improving the enantiopurity. The initial results indicate that while recrystallization can marginally enhance the ee, it has not yet achieved the level of 99%. We have added a section (section 10, Table S3) to the revised supplementary information.

Table S3. Recrystallization experiments of **Λ -28**

Solvent	First Recrystallization	Second Recrystallization
THF/petroleum ether	95% ee	95% ee
EtOAc/petroleum ether	94% ee	94% ee
DCM/petroleum ether	94% ee	95% ee

(5) Line 144 in main text, complex 26 should be complex 28; SI-page S77, the peak corresponding to the major enantiomer contains significant amount of impurity in the HPLC chart.

Response: As suggested, we have corrected the minor error in the main text. We have re-purified Λ -14 and included corresponding HPLC results in the revised supplementary information.

Reviewer #2 (Remarks to the Author):

In this manuscript, Ma and co-workers present a clever and elegant strategy to synthesize chiral-at-iridium complexes by kinetic resolution (kr) of the corresponding brominated racemic precursors through Pd-catalyzed asymmetric Suzuki-Miyaura cross-coupling. From a practical point of view, it is questionable whether this approach (requiring 10 mol% of a Pd source, 12 mol% of a chiral phosphoramidite ligand, manipulation in a glovebox, and column chromatography for purification) is comparable to other methodologies that allow for a straightforward synthesis of many chiral-at-iridium complexes (for instance the use of readily available resolution agents like L-proline or L-serine, following the well-established methodologies developed by Meggers). Nevertheless, there is no doubt that, conceptually, this catalytic approach is extraordinarily relevant. Additionally, it widens the available tools for the resolution of chiral-at-metal compounds, and it can become the Holy Grail for the resolution of other stereogenic-at-metal compounds for which the “classical” methodologies are not applicable, provided that the central core is configurationally stable.

The authors have demonstrated the feasibility of the approach and its practical application for synthesizing a wide variety of derivatives. For this purpose, they have explored the reactivity of the

formed product and the unreacted brominated substrate (both obtained as enantioenriched samples after kr), showing that a broad range of synthetic transformations are possible.

Considering the novelty and relevance of the strategy and its potential impact in the field, I strongly support its publication in Nature Communications.

Nevertheless, there are some suggestions and minor corrections that could help the authors improve the quality of the work.

Thank the reviewer for thorough review and constructive feedback. We appreciate the reviewer's recognition of the novelty of our kinetic resolution strategy for synthesizing chiral-at-iridium complexes. We are pleased to see the reviewer's strong support for publication in Nature Communications, given the potential impact of our work. We have carefully addressed suggestions and minor corrections to enhance the quality of our manuscript.

Suggestions:

- The work presented is thoroughly supported by many experimental evidences, and there is no doubt that kr occurs with more than acceptable selectivities on a wide range of substrates and with different coupling partners. Nevertheless, it would be nice to include some discussion or hypothesis on the origin of the observed selectivities. In the last section of the manuscript, the authors conduct some kinetic experiments to define the rate-determining step of the process. Since the reaction is run under kinetic resolution conditions (selectivity being controlled by the rate of formation of delta and lambda Suzuki-Miyaura adducts) and the authors identify reductive elimination as rds, do they assume that reductive elimination is also the selectivity-determining step?

Response: We appreciate the reviewer's insightful comments and agree that a discussion on the origin of the observed selectivity would enhance our manuscript. We have conducted kinetic experiments to identify the rate-determining step as reductive elimination. However, given the selectivity is controlled by the rate of formation of the Δ and Λ oxidation adducts, we hypothesize that oxidative addition is the selectivity-determining step. We have incorporated the following sentence into the revised manuscript.

"Collectively, these kinetic studies indicate the reductive elimination might serve as the turnover-limiting step, Nevertheless, given the selectivity is controlled by the rate of formation of the Δ and Λ oxidation adducts, we still hypothesize that oxidative addition is likely the selectivity-determining step."

- Authors claim that the reaction is catalyzed by a monometallic Pd complex (presumably containing one chiral phosphoramidite ligand according to the P/Pd ratio used). Did the authors consider performing some theoretical calculations to evaluate the relative energy of the different diastereomeric reaction intermediates and transition states?

Response: We appreciate the reviewer's suggestion and acknowledge the importance of understanding the reaction mechanism at a deeper level. We attempted to seek collaboration with our colleagues, but were informed that the theoretical calculations would require a significant amount of time and computational resources due to the complexity molecular models. We will consider this suggestion for future work to further elucidate the reaction mechanism. For the current study, we have focused on the experimental validation of the catalytic system and the observed stereoselectivity.

- The preliminary optimization process, explained in detail in the Supplementary Information, should be mentioned in the main text. Actually, looking at the information presented in the SI, the authors explored a much larger number of chiral ligands than those shown in Table 1 (though not the same labeling scheme is used). Surprisingly, the discussion about the influence of the ligand properties on the effectivity of the kr in the main text is limited to a very simple comment on the effect of some electronic properties. Considering that the best results in terms of kr have been obtained with phosphoramidite ligands displaying an atropisomeric binaphthyl (or related) unit and stereogenic carbon centers, and some diastereomers of the same ligand have been explored (i.e. L7(=LS17) and LS25), it would be interesting to analyze/discuss if the enantioselection is mainly dictated by the chiral descriptor of the binaphthyl unit or by the stereogenic carbon atoms and which are the matching/mismatching combinations of both chiral elements. For this discussion, it would be necessary to specify the chiral descriptor (δ or λ) of the major enantiomer of the Suzuki-Miyaura adduct formed. Could the authors consider incorporating this information (if available) to Figures S1-S3 and Table 1?

Response: We appreciate the reviewer's suggestion to delve deeper into the influence of ligand properties on the enantioselectivity. Specifically, we will discuss the role of the chiral descriptor of the binaphthyl unit and the stereogenic carbon atoms in more detail. The experimental results indicate that the axial chirality of the binaphthyl unit plays a dominant role in controlling the enantioselectivity of the reaction. We have incorporated the following sentence in the revised supplementary information (Section 2, Figure S2).

"Notably, the diastereomer LS25 of LS17, when participating in the catalysis of the reaction, exhibits poor enantioselectivity. Homochiral phosphoramidite ligands (R,R,R or S,S,S) are more suitable for this reaction. Additionally, the experimental results indicate that the axial chirality of the binaphthyl unit plays a dominant role in controlling the enantioselectivity of the reaction."

- In this vein, the authors decided to label the residual unreacted substrate λ and the Suzuki-Miyaura adducts δ , suggesting that these are the main isomers of the unreacted and formed complexes, respectively. This descriptor is confirmed in the case of compounds δ -11 and λ -10, by analysis of the X-ray structure of derivative 40. It is stated that other complex pairs were further assigned accordingly by CD analysis. It seems unclear how these assignments were performed. Did the authors assume that with ligand L9 the main adduct formed was δ -configured regardless the organometallic core and arylboronic acid used? Why only a selection of compounds is analyzed by CD?

Response: We acknowledge the reviewer's concern regarding the assignments of isomers. The absolute configuration of Δ -40 was determined by X-ray crystallographic; the other assignments were made based on comprehensive CD analysis, which was performed for all relevant complexes. We have included CD data of all chiral Ir-complexes in the revised supplementary information (Section 9).

Minor corrections/typo mistakes:

- Please specify the reaction conditions in Figures 2 and 4.

Response: As suggested, we have added the reaction conditions in the revised manuscript

- In Figure 4 the superscript a is used to indicate that the reaction with ArBF₃K was performed at 40 °C. Please use the same label for the reaction forming product 31. Since all the reactions were conducted to close to 50% conversions, please specify the reaction time for each entry.

Response: As suggested, we have added the reaction time in the Figure 2 and Figure 4.

- In page 7 (line 121 in pdf). Please, rephrase: “Due to the exist”

Response: As suggested, we have rephrased the sentence, the revised title is now: “Due to the presence of two substituents at the biaryl of Δ-31.....”

- In Figure 5 the n-pentyl substituent in Δ-11 is missing.

Response: Thank the reviewer for pointing out the issue in Figure 5. We have corrected the structure of Δ-11.

- In the kinetic experiments, please specify the concentrations of Pd₂(dba)₃ and ligand used. In the Supplementary Information (page S181) it reads “concentrations of cat. used: 0.00075 M, 0.0015 M, 0.00225 M, 0.003 M.”

Response: As suggested, we have included the exact concentrations of Pd₂(dba)₃ and ligand used in the revised supplementary information (Section 8).

- In the kinetic study, one of the aryl-borates is labeled as 2m in the text and in Figure 7d. This labeling is unclear as this coupling partner was not included in Figure 4. Additionally, in Figure 4 the borates were not labeled.

Response: Thank the reviewer for pointing out the issue, the aryl-borate 2m only used in kinetic experiment (Figure 7d). Moreover, we have provided detailed labeling for all aryl-borates in the supplementary information (Section 6), ensuring clarity and completeness of our data presentation.

Reviewer #3 (Remarks to the Author):

In this manuscript, Ma and coworkers disclosed the asymmetric synthesis of stereogenic-at-metal complexes, through an innovative kinetic resolution strategy under a Pd-catalyzed asymmetric Suzuki-Miyaura cross-coupling reaction. Conventionally, the synthesis of stereogenic-at-metal complexes would rely on stoichiometric amounts of chiral reagents, auxiliaries or counterions, which indeed works, but represents very old-school methods, encountering with low step- and atom economies. Asymmetric catalysis, an ideal synthetic tool, if works, would undoubtedly triggers the revolution in the field. As such, this work is significantly notable for its innovation, as well as the obtained high selectivity (mostly >90% ee and s factor >50) in producing optically active stereogenic-at-iridium complexes from racemic mixtures. Further, the authors also demonstrated the synthetic diversification of the obtained complexes, showing the synthetic routes to metallodrugs and chiral photocatalysts. A solid kinetic investigation was ultimately performed, suggesting that the reductive elimination is the rate-determining step over Suzuki coupling reactions. The authors have contributed a thorough and robust work in this manuscript.

Overall, this reviewer would like to suggest the publication of this work in Nature Communications after the following issues are addressed.

Thank the reviewer for the comprehensive and positive review. We are delighted to receive the reviewer's strong support for the publication of our manuscript in Nature Communications, given its innovative approach to asymmetric synthesis of stereogenic-at-metal complexes. We have carefully addressed the following issues what the reviewer has raised to ensure the robustness and clarity of our manuscript.

1) Please rephrase the title in which “stereogenic-at-metal” might be more specific and acceptable than molecules featuring metal-centered chirality.

Response: We have carefully considered the reviewer's recommendation and have rephrased the title. The revised title is now: Asymmetric Synthesis of Stereogenic-at-Iridium(III) Complexes through Pd-Catalyzed Kinetic Resolution.

2) It's interesting to know the Rh-complex is more kinetically unstable. The authors may emphasize the configurational stability of Ir congeners towards heat, moisture, various solvents. Would be nice to know for any users.

Response: As suggested, we conducted stability tests of the compound **A-28** in various solvents. The results indicate that compound **A-28** maintains its enantioselectivity for 6 hours at 60°C in THF/H₂O(9/1), *i*-PrOH, or MeCN. We have added a section (section 11, Table S4) to the revised supplementary information.

Table S4. Configuration Stability Test of **A-28**

Solvent	25 °C (6 h)	60 °C (6 h)
THF/H ₂ O (9/1)	94% ee	94% ee
i PrOH	94% ee	94% ee
MeCN	94% ee	94% ee

3) Figure 6, the numbering of reaction conditions should start from 1) but not 2).

Response: Thank the reviewer for pointing out the issue with the numbering in Figure 6. We have corrected the numbering of the reaction conditions to start from 1) as suggested.

4) Please double check the minus symbol for compound 43 as well as others.

Response: As suggested, the compound 43 and other compounds has been double-checked and confirmed to be correct.

5) Sample concentration of the CD analysis must be provided in the figure legend.

Response: As suggested, we have included sample concentration of the CD analysis in the figure legend.